# Salivary-Gland-Mediated Nitrate Recirculation as a Modulator for Cardiovascular Diseases

**DOI:** 10.3390/biom15030439

**Published:** 2025-03-19

**Authors:** Baoxing Pang, Xingyun Qi, Huiliang Zhang

**Affiliations:** 1Department of Medical Genetics and Molecular Biochemistry, Lewis Katz School of Medicine, Temple University, Philadelphia, PA 19140, USA; 2Department of Biology, Rutgers University, Camden, NJ 08103, USA

**Keywords:** cardiovascular disease, nitrate, nitrite, nitric oxide (NO), nitrate–nitrite–NO pathway, salivary gland

## Abstract

Cardiovascular diseases (CVDs), which include multiple disorders of the heart and blood vessels, are the leading causes of death. Nitric oxide (NO) is a vasodilator that regulates vascular tension. Endogenous NO is produced via the L-arginine–nitric oxide synthase (NOS) pathway. In conditions of cardiovascular dysfunction, NOS activity is impaired, leading to NO deficiency. In turn, the reduction in NO bioactivity exacerbates the pathogenesis of CVDs. Exogenous intake of inorganic nitrate supplements endogenous production via the nitrate–nitrite–NO pathway to maintain the NO supply. Salivary glands play an essential role in the conversion of nitrate to NO, with approximately 25% of circulating nitrate being absorbed and secreted into saliva. As a result, salivary nitrate concentrations can exceed that in the blood by more than tenfold. This recycled nitrate in saliva serves as a reservoir for NO and performs NO-like functions when endogenous NO production is insufficient. In this review, we summarize the emerging benefits of dietary nitrate in CVDs, with a particular focus on salivary-gland-mediated nitrate recirculation in maintaining NO bioavailability and cardiovascular homeostasis. Salivary-gland-mediated nitrate recirculation provides a novel perspective for potential intervention of CVDs.

## 1. Introduction

Cardiovascular diseases (CVDs) encompassing a wide range of cardiac and vascular disorders are the leading causes of mortality worldwide [1]. The pathological process of CVDs is influenced by elevated blood pressure, endothelial dysfunction, and platelet aggregation [2]. Reductions in nitric oxide (NO) production and bioavailability are among the main features and causes of CVDs [3]. Almost half a century ago, NO was identified as a vasodilator in the endothelial cells [4]. This discovery uncovered the mechanism of nitroglycerin, the first organic NO donor, on dilating the vessels to improve the heart blood supply and release angina [5]. Later on, NO was revealed to play multiple roles in the cardiovascular system homeostasis by sustaining blood flow, regulating platelet aggregation and leukocyte adhesion, and protecting endothelial integrity [6,7,8]. Notably, NO deficiency is one of the most common pathological causes of CVDs [9]. A sufficient and stable NO supply is required to maintain cardiovascular function, and restoring NO homeostasis is a potential therapeutic option for CVDs [10].

NO can be produced through two pathways: (1) the L-arginine–NO pathway, which relies on nitric oxide synthases (NOSs), including endothelial NOS (eNOS), neuronal NOSs (nNOSs), and inducible NOSs (iNOSs); and (2) the nitrate–nitrite–NO pathway, which involves the conversion of the dietary inorganic nitrate to nitrite, primarily by oral bacteria, followed by the reduction of nitrite into NO in the stomach acidic environment [11]. The nitrate–nitrite–NO pathway is a major contributor to NO generation, serving as an important alternative to the classic NOS-dependent pathway [12]. The generation, regulation, and physiological and pathophysiological roles in the cardiovascular system of the NOS-dependent pathway have been extensively investigated and summarized [13,14,15,16]. In this review, we specifically focus on the nitrate–nitrite–NO pathway in the context of CVDs.

Inorganic nitrate (NO_3_^−^), a NO donor, is commonly found in green vegetables. In 2008, Dr. Lundberg initially proposed the importance of the nitrate–nitrite–NO pathway in the human body, and he pointed out that dietary nitrate could play a positive role in many pathological conditions such as CVDs, hypertension, and gastric ulcers [17]. The cardiovascular system benefits from a nitrate-rich vegetable diet that complements the classic L-arginine–NOS pathway [18]. Supporting this notion, Dr. Bondonno highlighted that vegetable-derived nitrate acts as an important cardioprotective chemical resource through its sustainable NO generation [19].

Interestingly, salivary glands concentrate plasma nitrate, which raises salivary nitrate concentrations by 10–20 times that in the blood [20]. Then, concentrated nitrate is secreted into saliva by the salivary glands, where oral bacteria transform it into nitrite, providing sustainable and reliable NO resources. The contributions of the nitrate–nitrite–NO pathway to physiology and disease have been summarized [21,22]. However, several important questions need further discussion: What is the significance of nitrate recirculation mediated by the salivary glands? What are the roles of salivary glands and oral bacteria in regulating NO generation? Why is vegetal nitrate considered a relatively safe source for NO production? To explore these questions, in this review, we summarize salivary-gland-mediated nitrate recirculation and the importance of NO homeostasis regulated by the enterosalivary nitrate–nitrite–NO pathway, with an emphasis on CVDs.

## 2. Vegetal Nitrate Source

Inorganic nitrate is an essential element of the natural nitrogen cycle [23]. Human diets, especially green vegetables, are significant sources of dietary nitrate [24]. Dietary nitrate has been regarded as an unhealthy NO oxidation product for more than 50 years, with worries about its possible carcinogenic implications because of the nitrogen compounds it may form [25]. However, recent clinical evidence indicates that dietary nitrate, particularly from vegetables, is well absorbed and recycled and supports many physiological functions [26,27].

Green vegetables are the main source of dietary nitrate intake. While vegetables such as broccoli, cucumber, onions, and potatoes have relatively low nitrate amounts, leafy greens including spinach, beetroot, lettuce, cress, rucola, and celery have very high nitrate levels [28,29,30]. In general, nitrate concentrations are higher in leafy vegetables than in seeds and tubers. We divide common vegetables into four groups according to their nitrogen content to make it easier to grasp their nitrate content level. Specifically, the vegetables that are extremely high in nitrate (˃2500 mg/kg) include spinach, beetroot, and celery; those that are high in nitrate (1000–2000 mg/kg) include parsley and leek; those that are medium in nitrate (500–1000 mg/kg) include cabbage and turnip; and those that are low in nitrate (˂500 mg/kg) include broccoli and cucumber (Figure 1).

After ingestion, generally, the plasma nitrate levels peak 15 to 30 min following a nitrate-rich meal, and with a half-life of 5 to 10 h in the body [31]. However, both the peak time and elimination half-life of nitrate may vary depending on the dose and type of nitrate-rich food consumed. For instance, in a beetroot juice nitrate formula clinical trial, blood nitrate can peak from 1 to 3 h, with a half-life ranging from 7 to 26 h [32]. The salivary glands, kidneys, and biliary system resorb around 25% of nitrate in the blood, leaving the rest of the nitrate excreted by the urine [33]. The exogenous nitrate–nitrite–NO pathway becomes much more active under hypoxic and ischemic conditions in the body, whereas the endogenous L-arginine–NOS pathway is blocked to a certain extent [34]. Under these stress conditions, dietary nitrate restores NO bioavailability as an efficient NO donor.

## 3. The Salivary-Gland-Mediated Recirculation of Nitrate

The conversion and recirculation of the nitrate–nitrite–NO pathway depends heavily on the salivary glands. It is worth noting that salivary glands actively collect and concentrate circulating nitrate and store it in saliva as a backup. Oral commensal bacteria convert salivary nitrate to nitrite; then, the stomach transforms the nitrite into NO [35]. As a highly reactive and short-lived molecule, NO exists in its free form with a half-life only of 2–6 s. However, NO can react with thiols in the cysteine residues, forming S-nitrosothiols (SNOs). Notably, it can bind to the heme moiety of hemoglobin in the plasma, resulting in the formation of S-nitrosohemoglobin (SNO-Hb), which might be a reservoir of the NO [36].

Sialin, a nitrate transporter in salivary glandular epithelial cells, uptakes and concentrates the nitrate [37]. The nitrate absorption from the blood at the salivary gland enhances nitrate circulation and preserves the NO balance. Saliva, which is composed of organic materials, dissolved minerals, and 99.5% water, is essential for maintaining oral and general health [38]. Human salivary glands generate between 0.5 and 1.5 L of saliva each day [39]. In a clinical trial, after nitrate supplementation, plasma nitrite levels had no increase in individuals who did not swallow saliva, emphasizing the value of nitrate recirculation mediated by the salivary glands [40]. Thus, it is reasonable to believe that conditions or drugs that inhibit salivary flow could reduce nitrate recirculation and, consequently, the NO storage pool.

Oral nitrate-reducing bacteria are also critical for the transformation of dietary nitrate. As an entrance to the human body, the mouth cavity contains one of the five major microbial ecosystems, and is intimately related to cardiovascular and metabolic disorders [41]. Because humans and other mammals lack a specific and efficient nitrate reductase, oral microbial populations are indispensable for the reduction of nitrate to nitrite [42]. It has been reported that antibacterial mouthwash drastically affects the composition of oral bacteria, resulting in a lower bioavailability of NO [43]. These findings highlight the critical role of oral bacteria in NO production from nitrate. Collectively, dietary inorganic nitrate, processed through enterosalivary circulation, provides a reliable and safe strategy for maintaining NO levels. The exogenous intake of inorganic nitrate and salivary-gland-mediated nitrate recirculation are illustrated in Figure 2.

## 4. Protection and Mechanisms of Inorganic Nitrate on Cardiovascular Diseases

Through the salivary-gland-mediated nitrate–nitrite–NO pathway, inorganic nitrate supplementation, especially from nitrate-rich vegetables, is an effective cardioprotective strategy. Dietary nitrate has been shown in multiple clinical trials to have considerable biological NO-like effects, including reducing blood pressure and enhancing cardiac, endothelial, and ischemia-reperfusion damage [44,45].

The mechanism of the nitrate–nitrite–NO pathway likely overlaps with the L-arginine–NO pathway, as both ultimately produce NO. NO activates the soluble guanylyl cyclase (sGC) to form cyclic guanosine monophosphate (cGMP), which in turn activates protein kinase G (PKG) and phosphodiesterase (PDE) [46]. The activation of the PKG pathway modulates calcium handling by regulating L-type calcium channels (LTCC), large-conductance Ca^2+^-activated K^+^ channels (BK channels), phospholamban (PLB), and the inositol trisphosphate receptor (IP3R) [47]. Additionally, the PKG pathway activates myriad transcription factors to regulate gene expression. Another key effect of NO is the post-translational modification S-nitrosylation, through which a nitrosyl group is added to the thiol group of cysteine to form S-nitrosothiol (SNO) [48]. NO also forms reactive nitrogen species (RNS), such as nitroxyl anion, to play pleiotropic roles by posttranslational modifications and interactions with reactive oxygen species (ROS) [49] (see more details in Section 4.6).

It is important to mention that a nitrate-rich diet, particularly a vegetal diet, may offer benefits beyond the nitrate–nitrite–NO pathway. Other ingredients, such as vitamin C, can enhance the role of nitrate [50]. Furthermore, modulation of oral microbiota may provide an efficient strategy to boost the beneficial effects of the enterosalivary nitrate–nitrite–NO pathway.

### 4.1. Blood Pressure Lowered by Nitrate-Rich Vegetables

Hypertension is a common disorder that causes significant morbidity and mortality and is also a major risk factor for CVDs. NO is widely recognized as a vital signaling molecule involved in a variety of cardiovascular functions, including blood vessel tone, immunological function, and related metabolic management [51]. NO is also an important vasodilator that regulates vascular homeostasis, and a decrease in NO bioactivity is associated with hypertension [52]. Inorganic nitrate, which can be obtained from sources such as sodium nitrate, potassium nitrate, beetroot juice, or nitrate-rich vegetables, has been proven to have both short- and long-term blood-pressure-lowering benefits in rats and humans [53,54,55,56,57]. For the first time, Larsen et al. found that when healthy participants consumed sodium nitrate, their diastolic blood pressure (DBP) decreased by 3.7 mmHg [58]. Further studies verified that dietary nitrate, from beetroot juice or vegetables high in nitrate, lowered DBP by 5.3 mmHg [59]. One month of intervention with beetroot juice significantly decreased DBP by 2.4 mmHg in patients with hypertension [60]. Furthermore, the blood-pressure-lowering benefits of inorganic nitrate persist during and after exercise [61]. A nitrate-rich vegetable diet has been shown to lower blood pressure in healthy individuals, prehypertensive groups, and even patients with hypertension receiving medication treatment [62]. Notably, no adverse or side effects were observed in the clinical trials employing nitrate-rich vegetables.

### 4.2. Nitrate Supplementation Attenuates Heart Failure

There is growing evidence that nitrate efficiently improves heart performance in patients and animals with heart failure. Inorganic nitrate alleviates angiotensin-II-induced cardiac hypertrophy, endothelial dysfunction, and cardiac fibrosis [63]. Thirteen days of beetroot juice supplementation containing inorganic nitrate improved submaximal exercise test results in males with heart failure who had a lower ejection fraction [64]. In addition, a recent study showed that dietary nitrate improved glucose intolerance, reduced the production of reactive oxygen species (ROS), alleviated intestinal dysbiosis, and attenuated cardiac remodeling in an obese mice model [65]. The cardioprotective benefits of nitrate were found in this study to be mainly associated with decreasing the level of gut dysbiosis rather than lowering blood pressure [65]. This result implies that the management of intestinal microbiota homeostasis by nitrate may contribute to the improvement of cardiac remodeling. In animals with heart failure, sodium nitrite has been demonstrated to restore NO-mediated cytoprotective signaling, improving left ventricular performance of the heart [66]. In conclusion, it was found that both dietary nitrate and nitrite reduce cardiac remodeling in heart failure.

### 4.3. Ischemia-Reperfusion Injury Alleviated by Nitrate–Nitrite–NO Pathway

Tissue ischemia-reperfusion (I/R) injury occurs when blood flow restored after a period of ischemia or hypoxia. I/R injury to the heart is one of the leading causes of mortality and morbidity [67]. Myocardial I/R injury is caused by a variety of factors, particularly oxidative stress, proinflammatory activation, and vascular endothelial dysfunction [68]. In particular, reduced NO bioavailability exacerbates myocardial injury and is an important factor in the pathogenesis of I/R injury [69]. One possible treatment for I/R injury is nitrate therapy, which increases NO levels in ischemic tissues. Nitrate or nitrite has shown protective effects against I/R injury in the heart [70], brain [71], liver [72], and hind-limb [73]. In a rat heart I/R model, the treatment with the NO donor S-nitroso-N-acetyl-DL-penicillamine (SNAP) improved systolic, diastolic, and coronary reperfusion function [74]. Another NO donor, sodium nitroprusside, also enhanced rat heart function during ex vivo I/R [75]. I/R heart function improvement was also reported in mice, cats, and dogs by administering nitrate/nitrite before, during, and after the ischemia [76,77,78]. Our group also found that oral sodium nitrate supplements mitigated skin flap I/R injury through reduced oxidative stress and inflammatory responses [79].

The vasodilatory increase in blood flow mediated by the nitrate–nitrite–NO pathway is probably responsible for nitrate/nitrite protection in ischemic tissue reperfusion. Dietary nitrate exerts its vasodilatory effects primarily through vascular smooth muscle relaxation by reduction to NO, which in turn promotes the typical NO-cyclic guanylate monophosphate (cGMP)-protein kinase G signaling cascade [80]. In addition, dietary nitrate has been shown to increase blood flow in various vascular beds, which benefits the perfusion of the cardiovascular tissue. It was found that the brain, stomach, and skin tissues all benefit from increased blood flow when dietary nitrate is consumed [81,82,83,84,85]. A clinical study showed that nitrates promote vasodilation, which increases blood flow to the severely hypoperfused heart [86]. The increased bioavailability of NO provided by nitrate therapy enhances the vasodilatory effect to reduce ischemia and hypoxia, especially in special hypoxic or acidic environments.

### 4.4. Endothelial Dysfunction Improved by Inorganic Nitrate

The endothelium plays an essential role in vascular homeostasis, and endothelial dysfunction is strongly associated with the development of heart failure, high systemic vascular resistance, and hypertension [87]. One of the main causes of compromised endothelium-dependent vasodilation and contraction is an insufficient supply of NO [88]. NO donors can regulate endothelial dysfunction. Nitrate-rich spinach improves endothelial function in healthy individuals [89]. Furthermore, oral nitrate supplementation improves endothelial function and walking capacity in individuals with peripheral artery disease [90]. Age, increasing obesity, and elevated systolic blood pressure may reduce the beneficial effects of inorganic nitrate on endothelial function [91].

Platelets, which play a key role in acute thrombotic events, also contribute to the pathophysiology of endothelial dysfunction. NO primarily regulates platelet adhesion to the endothelium and platelet aggregation. Anticoagulant activity is reduced because of insufficient NO levels, leading to endothelial dysfunction. Chronic eNOS deficiency decreases NO availability, resulting in reduced platelet aggregation [92]. According to a study by Webb et al., dietary nitrate effectively prevented endothelial dysfunction in the human forearm caused by an ischemic insult and further decreased platelet aggregation [54]. Dietary nitrate may also reduce platelet aggregation during endothelial dysfunction, as inorganic nitrate has an antiplatelet effect in animal models of eNOS impairment. This is supported by a clinical trial showing that nitrate supplementation reduced platelet-derived extracellular vesicles in patients with coronary artery disease receiving clopidogrel [93]. These findings suggest that dietary nitrate may reduce the risk of thrombus formation in patients with CVDs.

In addition, NO is essential for leukocyte adhesion and chronic inflammation associated with endothelial dysfunction. Dietary nitrate reduces a variety of blood-soluble inflammatory markers in middle-aged or older individuals with hypertension. Accordingly, a recent study showed that inorganic nitrate, by increasing NO levels, partially corrected the endothelial dysfunction caused by periodontitis [94]. In addition, our research confirmed that dietary nitrate reduced inflammatory factors and accelerated the healing of infected skin wounds in an animal model [81]. In conclusion, inorganic nitrate supplementation may enhance endothelial function by serving as a reservoir of NO.

### 4.5. Mitochondrial Reactive Oxygen Species (ROS) Inhibited by Dietary Nitrate

Mitochondria, as the primary source of ROS generation, are also modulated by NO. Dietary nitrate reduced lipid accumulation, mitochondrial ROS production, and oxidative stress markers in the liver of a high-fat-diet-induced animal model [95]. Nitrate supplementation also protects mitochondrial complex I and maintains mitochondrial energetics in cardiac hypoxia, indicating that dietary nitrate is a promising treatment to preserve mitochondrial function and bioenergetics [96]. Additionally, in a short-term (three-day) skeletal muscle disuse model, dietary nitrate arrested the decline in mitochondrial protein synthesis rate, mitochondrial content, and bioenergetics [97]. Nitrate also directly protects the mitochondrial function in the cell model. Lizheng Qin et al. found that xanthine oxidoreductase-derived NO maintained mitochondrial function and partially suppressed lipopolysaccharide-induced inflammation in mouse RAW 264.7 macrophages [98]. Nitrate therapy can reduce mitochondrial ROS production and increase mitochondrial efficiency.

### 4.6. Reactive Nitrogen Species (RNS) and Cardiac Contractility

Reactive nitrogen species (RNS) are a group of highly reactive nitrogen-containing molecules derived from nitric oxide. RNS include principal nitroxyl anion, nitrosonium cation, higher oxides of nitrogen, S-nitrosothiols, and dinitrosyl iron complexes, among others [49]. RNS increases Ca^2+^ sparks and ryanodine receptor (RyR) sensitivity to enhance calcium transient and contractility [99]. Nitroxyl, a one-electron reduction of nitroxyl anion generated from Angeli’s salt, enhances cardiac function with a positive inotropy effect in the normal heart [100] and failing heart [101]. Notably, this inotropic effect is independent of beta-adrenergic or cGMP signaling. Nitroxyl increases RyR open probability and ATP-dependent Ca^2+^ uptake activity [102]. Another mechanism of nitroxyl enhancing the contractile function is mediating the disulfide bond formation between tropomyosin (Cys190) and actin (Cys257) and the disulfide bond between myosin heavy chain and myosin light chain 1 [103]. In addition, the NO donor S-nitroso-N-acetylpenicillamine (SNAP) can reversibly increase Ca^2+^ sparks without mechanical stretching [104].

Given the wide-ranging preventive function of dietary nitrate in cardiovascular homeostasis, we summarized its potential therapeutic benefits and underlying mechanisms in CVDs in Figure 3.

## 5. Safety of Nitrate-Rich Diet

Nitrate recirculation, mediated by the salivary glands, is an essential NO reservoir. As a green NO donor, nitrate as a supplement improves cardiovascular health. Meanwhile, concerns have been raised regarding the potential adverse effects that prolonged exposure to nitrates may cause, particularly an increased risk of cancer. It has been known for many years that the conversion of nitrate and nitrite to nitrosamines poses a carcinogenic threat [105].

Current evidence suggests that dietary nitrate supplements, particularly those derived from nitrate-rich vegetables, are unlikely to pose significant health risks [106,107,108,109]. Nitrate-rich vegetables are considered safe NO donors in contrast to cured meats, which have increased the risk of CVDs [110]. It is generally accepted that green vegetables contain high levels of nitrate and many healthy components, such as dietary fiber, vitamin C, carotenoids, polyphenols, and other elements that may promote cardiovascular health. More significantly, a study by Lundberg et al. showed that dietary nitrite prolonged longevity and protected fruit flies against age-related locomotor decline, with no negative effects [111]. In a follow-up study, they found that in an aging rat model, nitrate supplementation improved the function of the endothelium and prolonged the lifespan of rats [112]. Thus, modest dietary nitrate intake, particularly from nitrate-rich vegetables, is a safe and valuable NO donor.

## 6. Conclusions

In conclusion, the enterosalivary nitrate–nitrite–NO pathway improves mitochondrial activity, regulates Ca^2+^ handling, lowers blood pressure, protects against heart failure and ischemia-reperfusion injury, and preserves endothelial cell function. Nitrate-rich vegetables, which are readily available, provide economical and effective treatment for NO deficiency. Specifically, through the enterosalivary circulation, dietary inorganic nitrate is stored in saliva, providing a sustainable approach to restore NO levels and a great benefit to CVDs. This review highlights the importance of nitrate recirculation mediated by the salivary glands and calls attention to the relationship among the salivary glands, oral bacteria, and the NO homeostasis in the body.

## 7. Perspectives

Several important aspects of salivary-gland-mediated nitrate recirculation and its protective role in CVDs require further investigation. First, many factors may decrease salivary production and impair nitrate recirculation, including radiation-induced damage to the parotid gland, oral bacterial homeostasis during head and neck cancer treatment, autoimmune diseases such as Sjögren’s syndrome, and many medications that may decrease salivary production [113]. In these cases, reduced salivary flow could lead to a deficiency in NO bioavailability via the nitrate–nitrite–NO pathway, potentially disrupting cardiovascular homeostasis. Second, the physiological functions and underlying mechanisms of supplemental nitrate delivery need to be investigated. Specifically, whether and how does nitrate influence other systems, such as the nervous system and immune system, and in turn regulate cardiac and vascular function? Third, it is important to determine a safe level of nitrate supplementation. Establishing a clear guideline for nitrate intake is critical for both clinical management and everyday dietary recommendations.

## Figures and Tables

**Figure 1 biomolecules-15-00439-f001:**
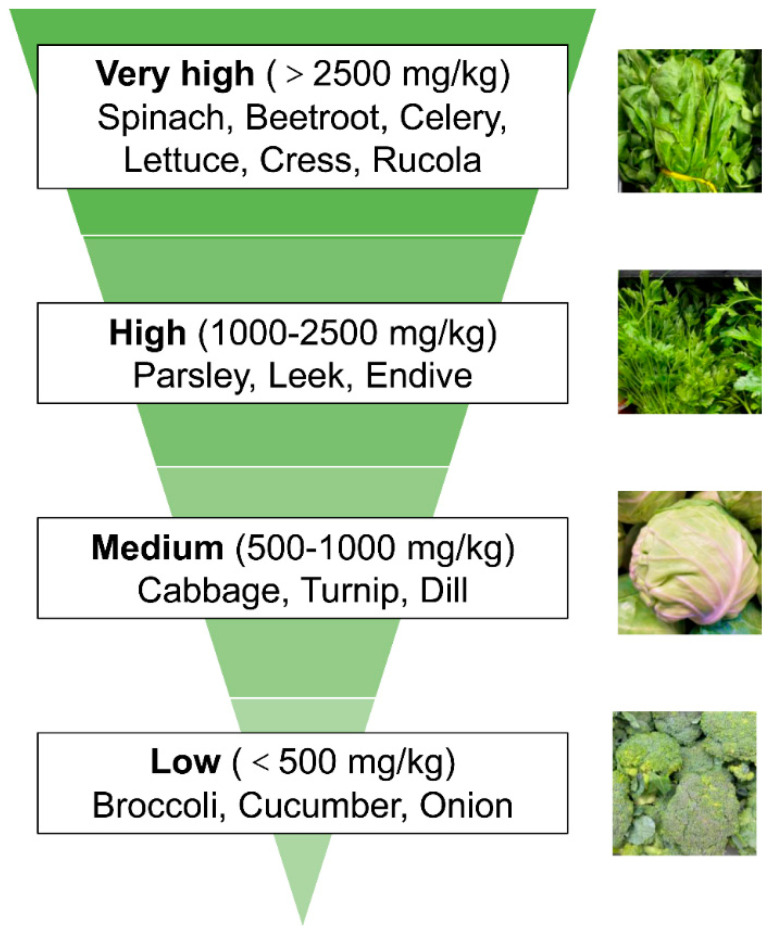
Classification of common dietary vegetables based on their nitrate content. Nitrate levels in dietary vegetables vary from very high level to low level. Specifically, vegetables with very high nitrate levels (˃2500 mg/kg) include spinach, beetroot, and celery; those with high nitrate levels (1000–2000 mg/kg) include parsley and leek; vegetables with medium nitrate levels (500–1000 mg/kg) include cabbage and turnip; and vegetables with low nitrate levels (˂500 mg/kg) include broccoli and cucumber.

**Figure 2 biomolecules-15-00439-f002:**
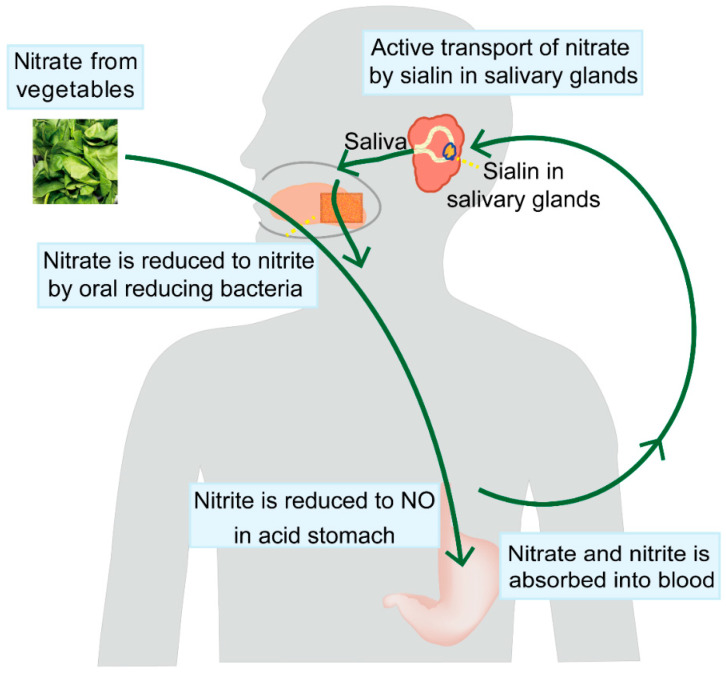
Salivary-gland-mediated recirculation of nitrate. To maintain the NO reservoir, the salivary glands actively absorb up to 25% of the circulating nitrate and concentrate it more than ten times in saliva. Sialin, a nitrate transporter in the parotid gland, mediates this nitrate recirculation. Nitrate-reducing bacteria in the mouth convert nitrate to nitrite when nitrate-rich saliva is released into the oral cavity. While most of the remaining nitrate and nitrite are absorbed into the systemic circulation, a portion of the nitrite is subsequently transformed into NO in the stomach.

**Figure 3 biomolecules-15-00439-f003:**
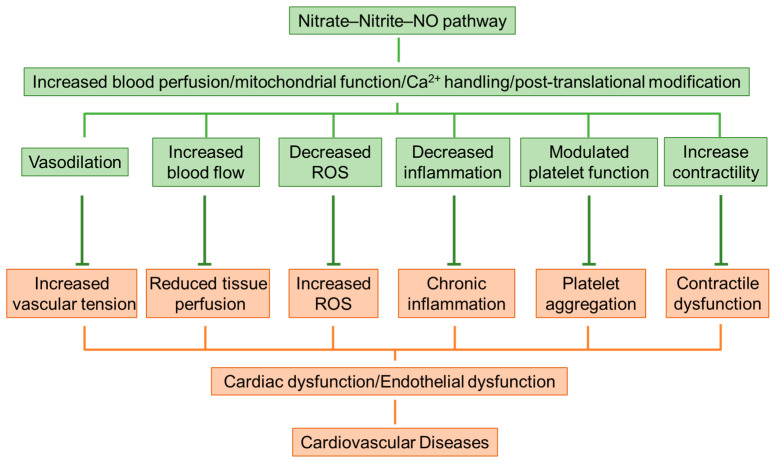
Potential therapeutic benefits and underlying mechanisms of inorganic nitrate in cardiovascular diseases. Orange highlights CVDs, including reduced tissue perfusion, increased vascular tension, and increased production of ROS. Green indicates potential therapeutic effects and mechanisms. Nitrate therapy may decrease the progression of cardiovascular disease by improving platelet function, lowering mitochondrial ROS, increasing blood perfusion, and providing anti-inflammatory benefits.

## Data Availability

Not applicable.

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
