# Peer review of "Salivary-Gland-Mediated Nitrate Recirculation as a Modulator for Cardiovascular Diseases"

_biomolecules, 2025, doi:10.3390/biom15030439_

Round 1

Reviewer 1 Report

Comments and Suggestions for Authors

This study by Dr. Pang describes the emerging benefits of dietary nitrate in cardiovascular diseases focusing on salivary gland-mediated nitrate recirculation in maintaining NO bioavailability and cardiovascular homeostasis. The author summarizes the role of diet-mediated enterosalivary nitrate-nitrite-NO pathway as a novel perspective on potential intervention of cardiovascular disease. Overall, although it is well organized, I think it would be better if there was a little more consideration from a molecular biology perspective.

I have a couple of questions。

(1) How do you think the dietary enterosalivary nitrate-nitrite-NO pathway ultimately exerts NO bioavailability in the myocardium? Is it through NO itself or transnitros(yl)ation? And is the carrier red blood cells or plasma proteins? I would appreciate it if you could explain as much as you know in the text.

(2) In mitochondria-rich tissues such as cardiac muscle, not only reactive oxygen species but also reactive nitrogen species (peroxynitrite from excessive NO generation) are generated and can cause myocardial damage. Can you explain why dietary (exogenous) nitrate has myocardial protection effect?

(3) Can you provide some additional evidence on factors that further enhance the beneficial effects of the enterosalivary nitrate-nitrite-NO pathway on the cardiovascular system (e.g. foods including reducing agents, gastric acid secretion, oral microorganisms, etc.)?

Author Response

This study by Dr. Pang describes the emerging benefits of dietary nitrate in cardiovascular diseases focusing on salivary gland-mediated nitrate recirculation in maintaining NO bioavailability and cardiovascular homeostasis. The author summarizes the role of diet-mediated enterosalivary nitrate-nitrite-NO pathway as a novel perspective on potential intervention of cardiovascular disease. Overall, although it is well organized, I think it would be better if there was a little more consideration from a molecular biology perspective.

Response: We appreciate the reviewer’s positive comments. In this revision, we incorporated several paragraphs discussing the molecular mechanisms of nitrate-nitrite-NO. Please refer to the detailed responses to each question for further information.

I have a couple of questions.

  • How do you think the dietary enterosalivary nitrate-nitrite-NO pathway ultimately exerts NO bioavailability in the myocardium? Is it through NO itself or transnitros(yl)ation? And is the carrier red blood cells or plasma proteins? I would appreciate it if you could explain as much as you know in the text.

Response: Thank you for this suggestion. NO is the final effective product in the nitrate-nitrite-NO pathway. We add the discussion “As a highly reactive and short-lived molecule, NO exists in its free form with a half-life only of 2 – 6 seconds. However, NO can react with thiols in the cysteine residues, forming S-nitrosothiols (SNOs). Notably, it can bind to the heme moiety of hemoglobin in the plasma, resulting in the formation of S-nitrosohemoglobin (SNO-Hb), which might be a reservoir of the NO [33].” See Section 3, lines 112-117.

  • In mitochondria-rich tissues such as cardiac muscle, not only reactive oxygen species but also reactive nitrogen species (peroxynitrite from excessive NO generation) are generated and can cause myocardial damage. Can you explain why dietary (exogenous) nitrate has myocardial protection effect?

Response: We agree with the reviewer that reactive nitrogen species (RNS) play a crucial role in cardiac function and diseases. In response, we added a new section 4.6 to discuss the functions of RNS (see lines 278-292). Also, we included a discussion on the mechanisms of nitrate protection in Section 4, lines 152-164.

  • Can you provide some additional evidence on factors that further enhance the beneficial effects of the enterosalivary nitrate-nitrite-NO pathway on the cardiovascular system (e.g. foods including reducing agents, gastric acid secretion, oral microorganisms, etc.)?

Response: Thank you for the suggestion. It has been reported that Vitamin C enhances the effect of nitrate. Additionally, modulating oral microorganisms may serve as an effective strategy to boost the beneficial outcomes of the enterosalivary nitrate-nitrite-NO pathway. We included this discussion in lines 165-169.

Reviewer 2 Report

Comments and Suggestions for Authors

The presented review focuses on vegetal nitrate source, salivary gland-mediated nitrate recirculation, and the importance of NO homeostasis regulated by nitrate in cardiovascular diseases, including blood pressure, heart ischemia-reperfusion injury, and endothelial dysfunction. The topic is interesting and the review is quite well written. I have only a few comments:

1) At the end of the Introduction, clearly state the scientific novelty of this review. Are there any other reviews on this topic and how does this review differ from them?

2) Design and methods of review must be described after Introduction.

3) The authors, with reference to Ref.28, provide values for the pharmacokinetics of exogenous nitrates. It is worth mentioning that there are more recent studies that confirm the nitrate absorption period indicated by the authors, but a slightly longer elimination half-life, up to 10 hours. There may be a dose-dependent effect on the pharmacokinetic curve. The type of food source of nitrate can also be influenced (Ref. 44 is also appropriately duplicated in section 2). This should be discussed further.

4) In the Conclusion, it is advisable to highlight the main postulates from the review that have scientific novelty. It is better to separate the perspectives into a separate section.

Author Response

The presented review focuses on vegetal nitrate source, salivary gland-mediated nitrate recirculation, and the importance of NO homeostasis regulated by nitrate in cardiovascular diseases, including blood pressure, heart ischemia-reperfusion injury, and endothelial dysfunction. The topic is interesting and the review is quite well written. I have only a few comments:

Response: Thank you for your kind appreciation.

  • At the end of the Introduction, clearly state the scientific novelty of this review. Are there any other reviews on this topic and how does this review differ from them?

Response: Thank you for your valuable advice. To highlight the novelty of this review, we added a new paragraph in the introduction (see lines 42-52) and expanded the last paragraph in the introduction (see lines 64-71).

  • Design and methods of review must be described after Introduction.

Response: We added the design with question guidance in the final paragraph of the introduction (see lines 64-71). Since this is a review paper, we do not have experiment methods to include. Thank you for your understanding.

  • The authors, with reference to Ref.28, provide values for the pharmacokinetics of exogenous nitrates. It is worth mentioning that there are more recent studies that confirm the nitrate absorption period indicated by the authors, but a slightly longer elimination half-life, up to 10 hours. There may be a dose-dependent effect on the pharmacokinetic curve. The type of food source of nitrate can also be influenced (Ref. 44 is also appropriately duplicated in section 2). This should be discussed further.

Response: We agree with the reviewer that the pharmacokinetics of exogenous nitrate is very important, and that the elimination half-life may be influenced by the dose and type of nitrate-rich food. We added a new discussion in the last paragraph of Section 2 (see lines 97-102)

4) In the Conclusion, it is advisable to highlight the main postulates from the review that have scientific novelty. It is better to separate the perspectives into a separate section.

Response: Thanks for your advice. We separated the Conclusion and the Perspectives sections. We also updated and highlighted the scientific novelty of this review in the Conclusion section (See lines 322-331).

Round 2

Reviewer 1 Report

Comments and Suggestions for Authors

The authors responded appropriately to my queries and revised the manuscript.

I'd like to accept the revised version.

Reviewer 2 Report

Comments and Suggestions for Authors

I approve revised version of paper.